

# How aerosol size matters in AOD assimilation and the optimization using Ångström exponent

Jianbing Jin[1,a], Bas Henzing[1,*], and Arjo Segers[1]

[1]TNO, Department of Climate, Air and Sustainability, The Netherlands
[a]Now at Jiangsu Key Laboratory of Atmospheric Environment Monitoring and Pollution Control, Jiangsu Collaborative Innovation Center of Atmospheric Environment and Equipment Technology, School of Environmental Science and Engineering, Nanjing University of Information Science and Technology, Nanjing, Jiangsu, China

**Correspondence:** Bas Henzing (bas.henzing@tno.nl)

**Abstract.** Satellite-based aerosol optical depth (AOD) has gained popularity as a powerful data source for calibrating aerosol models and correcting model errors through data assimilation. However, simulated airborne particle mass concentrations are not directly comparable to satellite-based AODs. For this, an AOD operator needs to be developed that can convert the simulated mass concentrations into model AODs. The AOD operator is most sensitive to the input of the particle size and chemical composition of aerosols. Furthermore, assumptions regarding particle size vary significantly amongst model AOD operators. More importantly, satellite retrieval algorithms rely on different size assumptions. Consequently, the differences between the simulations and observations do not always reflect the actual difference in aerosol amount.

In this study, the sensitivity of the AOD operator to aerosol properties has been explored. We conclude that to avoid inconsistencies between the AOD operator and retrieved properties, a common understanding of the particle size is required. Accordingly, we designed a hybrid assimilation methodology (*hybrid* AOD assimilation) that includes two sequentially-conducted procedures. First, aerosol size in the model operator has been brought closer to the assumption of the satellite retrieval algorithm via assimilation of Ångström exponents. This ensures that the model AOD operator is more consistent with the AOD retrieval. The second step in the methodology concerns optimization of aerosol mass concentrations through direct assimilation of AOD (*standard* AOD assimilation). The hybrid assimilation method is tested over the European domain using Moderate Resolution Imaging Spectroradiometer (MODIS) Deep Blue products. The corrections made to the model aerosol size information are validated through a comparison with the ground-based Aerosol Robotic Network (AERONET) optical product. The increments in surface aerosol mass concentration that occur either due to the *standard* AOD assimilation analysis or *hybrid* AOD assimilation analysis are evaluated against independent ground $PM_{2.5}$ observations. The *standard* analysis always results in relatively accurate posterior AOD distributions; however, the corrections are hardly transferred into better aerosol mass concentrations due to the uncertainty in the AOD operator. In contrast, the model AOD and mass concentration states are considerably more accurate when using the *hybrid* methodology.



# 1 Introduction

Aerosol transport models describe how particulate pollution is formed from precursor gasses, how pollution is transported by atmospheric dynamics, and how it is removed by chemical reactions or deposition. Transport models are an important part of the earth modeling system. Aerosol transport is relevant for understanding and predicting weather and climate because

aerosol particles redistribute energy through direct absorption and scattering of (solar) radiation and they serve as nuclei upon which cloud droplets and ice crystals form thus changing the cloud reflectivity (Twomey, 1977) and cloud lifetime (Albrecht, 1989). Aerosols also play a wider role in atmospheric chemistry and biogeochemical cycles in the Earth system (Andreae and Crutzen, 1997), for example by carrying nutrients to ocean ecosystems (Baker et al., 2003) or nitrogen deposition that significantly affects plant diversity (Bobbink et al., 2010). Air pollution is a major environmental risk to health and particulates

affect more people than any other pollutant. In their 2018 fact sheet, the World Health Organisation estimates air pollution is responsible for 4.2 million premature deaths worldwide in 2016 (World Health Organization, 2018), costing an estimated 5.7 trillion USD or 4.8% percent of global GDP (The World Bank, 2019). Air pollution levels continue to rise, the strongest in urban environments in low- and middle-income countries. Aerosol transport models contribute to understand life cycles of airborne aerosols and hence support effective emission reduction policies, and are therefore important elements of operational

air quality forecast and analysis systems. Despite the enormous importance and efforts to improve models, validation studies (Dennis et al., 2010; Zhang et al., 2012) continue to report inconsistencies with observations that have different origins, e.g. uncertain emission inventories (Fan et al., 2018), mismatch in transport (Solazzo et al., 2017), and removal procedures (Croft et al., 2012). This also implies that our understanding of how aerosol pollution will respond to mitigation strategies is still quite limited.

The introduction of the 2008 European Directive on Ambient Air Quality and Cleaner Air for Europe encouraged the use of model simulations to perform air quality management tasks such as air quality assessment, forecasting and planning (Europe Environmental Agency, 2011) that were previously performed using measurements. At the same time we see rapid advances in sensor technologies and the availability of aerosol measurements from large scale network activities that can complement the modelling activities. Those measurements are preferably used to calibrate models and to perform model error corrections by

application of data assimilation techniques (Kalnay, 2002). Examples of popular aerosol measurements used for this purpose are ground-based LIDAR data (Yumimoto et al., 2008), surface particular matter (PM) concentration observations (Lin et al., 2008; Jin et al., 2018, 2019a), polar-orbiting satellite observations (Schutgens et al., 2012; Khade et al., 2013; Yumimoto et al., 2016a; Di Tomaso et al., 2017; Jin et al., 2022), and geostationary remote sensing data (Yumimoto et al., 2016b; Jin et al., 2019b, 2020). Among available measurements, satellite aerosol products provide valuable information by their high spatial

coverage: a single instrument is used to observe a large spatial area making additional harmonisation efforts unnecessary.

Using satellite based observations to improve simulated aerosol concentrations is not straightforward. Aerosol model and remote sensing data are not comparable directly, since the model simulates aerosol mass concentrations while the sensor measures aerosol optical properties. In order to assimilate aerosol optical data, typically the 3D mass concentration fields are converted into 2D fields of the optical properties retrieved from the measurements. This conversion is performed by a model





operator that matches the retrieval algorithm. To obtain retrieved aerosol properties, assumptions need to be made about the aerosol type, size, and optical properties in order to obtain quantities such as aerosol optical depth (AOD). However, these assumptions may be inconsistent with similar assumptions in the AOD operator of the aerosol transport model that intends to assimilate the retrievals. At each assimilation cycle some of the difference between the transport model–derived AOD and the

retrieved AOD may simply be due to these inconsistent assumptions. Assimilating the observed radiance as done by Weaver et al. (2007) avoids this inconsistency issue. Using satellite reflectances (radiances) to improve surface concentrations (Drury et al., 2010) and aerosol emissions (Wang et al., 2012; Xu et al., 2013) are other examples of initiatives to avoid the mismatch between assumption in retrieval algorithms and model operators. However, although assimilation of radiances is promising to avoid inconsistencies, other scientific challenges remain. The most notable challenge might be that the information content

of space-borne radiance (Intensity) measurements is mostly limited to a few degrees of freedom (Veihelmann et al., 2007; Mishchenko et al., 1999; Tanré et al., 1996; Hasekamp and Landgraf, 2005). Consequently, additional *a priori* information is needed to simulate top-of-atmosphere radiance intensities, such as the solar spectrum, cloud handling, surface reflectance properties, meteorological information (temperature, pressure, humidity), and sun and satellite geometries, and all of these need to be known accurately.

To simulate AOD in a model, the observation operator relies on the chemical composition and aerosol size information from the transport model. Information on size is needed for other processes too, for example aerosol/gas-phase interaction and deposition (Khan and Perlinger, 2017). In transport models aerosol size distributions are represented by sectional (e.g., Jacobson (2001); Gong et al. (2003); Rodriguez and Dabdub (2004)) or modal (e.g., Ackermann et al. (1998)) approaches; the difference between these approaches has been reviewed by Zhang et al. (2002). In this study, the target application of the aerosol mod-

elling is to improve the forecast of particulate matter concentrations, where the aerosol size is not of major concern; therefore, a modal approach with a diameter-based parametrization is used. Assumptions on aerosol size are part of the parameterization, but these assumptions differ from those made in satellite retrieval algorithms. Harmonization of these assumptions is difficult, for example because aerosol retrieval algorithms differ per instruments, each with different assumptions on aerosol sizes and other parameters. It is therefore necessary to use different parameterizations of aerosol sizes in the model for each observation

operator that is used to simulate retrievals.

Information on the aerosol sizes could be obtained from measurements too. Satellite-based aerosol products are usually reported in several wavelength bands, and the multi-wavelength interpolated Ångström exponent (Ångström, 1929) actually contains aerosol size information. Up to now, this information is only occasionally used in aerosol studies (Schuster et al., 2006; Saide et al., 2013; Liu et al., 2019). Most of the AOD assimilation efforts only incorporate AOD observations at a single wave-

length into the model (referred to as *standard* AOD assimilation in the whole paper). A few studies assimilated remote sensing aerosol optical products at several wavelength bands, for example Schutgens et al. (2010) assimilated both the Ångström and AOD simultaneously, and Saide et al. (2013) assimilated multi-wavelength AODs. These two studies both assumed that the size-related mismatch between model and observations is due to the uncertainty in the distribution of aerosol emissions over the fine or coarse modes, and/or the distribution of aerosol mass over different particle types such as anthropogenic, mineral





dust, and sea salt. In addition, both assumed that the radii distribution of aerosol types is constant and known, which is is not the case for most aerosol mixtures.

In this study, we first explore the role that aerosol sizes play in the conversion of mass concentration to AOD. A common understanding of the role of the particle size in the AOD operator and the retrieval algorithms is necessary when trying to

calibrate the AOD computations by comparison with actual AOD observations. Aerosol extinction sensitivity has therefore been studied using an offline AOD operator code based on *Mie* scattering theory (De Rooij and Van der Stap, 1984). The sensitivity of the AOD calculations to aerosol sizes have been examined. When assimilating AOD observations it is important that assumptions on size are consistent between simulation and retrieval. Therefore, a *hybrid* AOD assimilation system has been designed that consists of two sequentially-conducted steps. The first step aims at estimation of the size distribution parameters

by assimilating Ångström exponents. The second step aims at estimating the aerosol mass distribution by assimilation of AOD, using the just estimated size distribution parameters. The *hybrid* assimilation has been compared to a reference that only assimilates AOD, which might still provide AOD fields that are in agreement with observations, but might lead to incorrect mass concentrations. To the best of our knowledge, this is the first time that Ångström assimilation has been coupled with a standard AOD assimilation to optimize the radius parameterizations of aerosol fields.

This paper is organized as follows: Section 2 illustrates the *Mie* theory-based AOD operator that converts aerosol mass concentrations into AOD values. By employing an offline AOD operator, aerosol extinction sensitivity experiments are conducted to explore the role of aerosol radius in the calculations. Section 3 describes the measurements that have been used in this study for assimilation or independent validation. Section 4 describes the LOTOS-EUROS chemical transport model (CTM) and the configuration used for this study. The *hybrid* assimilation methodology combining Ångström and AOD assimilation is

introduced in Section 5, and applied with observations from the MODIS satellite instrument. The increments in surface aerosol mass concentration induced by the assimilations are evaluated using $PM_{2.5}$ surface concentration measurements. Section 6 summarizes the results and discusses the added value of using *hybrid* AOD assimilation.

## 2   Computation of AOD from aerosol mass concentrations

The Aerosol Optical Depth (AOD) for a certain wavelength is a measure for the extinction of light by aerosols in the atmo-

sphere. Here we describe how AOD is usually computed in a simulation model, and specifically the role of the aerosol size distribution in this.

### 2.1   Aerosol species

When modelling aerosol concentrations, the aerosol types are often categorized into groups based on their chemical composition. In this study we distinguish five different types: black carbon, organic carbon, mineral dust, sea salt, and secondary

inorganic aerosol (sulphate, nitrate, and ammonium). For computation of AOD it is necessary to define the specific properties for each aerosol type. The *refractive index* is a complex number (with a real and an imaginary part) that quantifies the bending and attenuation of light by a layer of aerosols. The *hygroscopicity* describes the tendency of aerosols to absorb moisture from



**Table 1.** Mass density and hygroscopicity index for different aerosol types

| aerosol | inorganic aerosol (so4, nh3, no3) | black carbon (bc) | organic carbon (oc) | sea salt (ss) | dust |
|---|---|---|---|---|---|
| mass density (kg/m$^3$) | 1841 | 2000 | 2000 | 2165 | 2650 |
| hygroscopicity index $\mathcal{K}$ | 0.8 | 0.0 | 0.1 | 1.0 | 0.0 |

the surrounding atmosphere, which is important for the AOD since aerosol water impacts on scattering and absorption by altering aerosol size and refractive index. The hygroscopicity index and mass density for the five types of aerosols used in the following aerosol extinction sensitivity calculations and our aerosol model can be found in Table 1.

## 2.2 AOD operator

AOD is a direct measure of the total light lost in the atmospheric column, which occurs due to aerosol absorption and scattering along the radiation transmission path. It is, therefore, not directly comparable with simulated 3D aerosol mass concentrations, which is the state calculated in a simulation model. To perform model calibration through AOD observations, or to adjust the model using AOD assimilation, an AOD operator ($\mathcal{H}$) is necessary:

$$\boldsymbol{\tau^m} = \mathcal{H}\boldsymbol{X} \tag{1}$$

where $\boldsymbol{X}$ denotes the 3D aerosol mass concentration for all aerosol species, and $\boldsymbol{\tau^m}$ denotes the 2D field of simulated AOD values. Calibration or assimilation is then based the difference between the simulation and observations, refered to as the *innovation*:

$$\boldsymbol{d} = \boldsymbol{\tau^m} - \boldsymbol{\tau} \tag{2}$$

where $\boldsymbol{\tau}$ is the AOD measurement vector.

## 2.3 *Mie* theory

In most chemical transport models including the LOTOS-EUROS chemical transport model described in section 4.1, the numerical conversion from aerosol mass concentration into AOD simulation follows the *Mie* theory. The basis is to calculate the scattering and absorption coefficients of spherical particles with given radius and refractive index. In the *Mie* calculation, the model AOD $\tau^m$ is defined as a vertical integration of the extinction coefficient $\epsilon_{\text{ext}}$ (1/m) over $n$ model layers:

$$\tau^m = \sum_{k=1}^{n} \epsilon_{\text{ext}}^k \cdot z^k \tag{3}$$

where $\epsilon_{\text{ext}}^k$ and $z^k$ denote the extinction coefficient and layer thickness at the k$^{th}$ layer, and $\epsilon_{\text{ext}}$ is the product of the dimensionless extinction efficiency $\mathcal{Q}_{\text{ext}}$, the total cross section per unit mass $\mathcal{S}$ (m$^2$/g) and the aerosol mass concentration $\mathcal{C}$ (g/m$^3$):

$$\epsilon_{\text{ext}} = \mathcal{Q}_{\text{ext}} \cdot \mathcal{S} \cdot \mathcal{C} \tag{4}$$





**Table 2.** Control variables in the aerosol extinction sensitivity experiments

| Variables | Descriptions |
|---|---|
| aerosol species | sulfate aerosol (so4), black carbon (bc), organic carbon (oc), dust and sea salt (ss) aerosol |
| mean geometric radius | 10 nm to 4 $\mu$m |
| geometric standard deviation | 1.59 $\mu$m |
| relative humidity (RH) | 0, 0.8 |
| incident wavelength ($\lambda$) | 440, 470, 550, 650, 870 nm |

in here $\mathcal{Q}_{\text{ext}}$ equals the sum of scattering efficiency and absorption efficiency. It depends on the ratio of aerosol radius and incident wavelength, and the chemical composition (Hulst and van de Hulst, 1981). $\mathcal{S}$ itself is governed by the particle size and aerosol mass density. Their complex manners will be discussed later in Section 2.4.

Ångström exponent $\mathcal{A}$ has been introduced for measuring the variability of wavelength dependent extinction coefficients at

different incident wavelengths. $\mathcal{A}$ is a quantitative indicator of aerosol size (Ångström, 1929), specifically, it reflects the size of aerosols with sub-micrometer radius (O'Neill et al., 2001). Mathematically, $\mathcal{A}$ is the slope of the line from the AODs ($\tau_i$, $\tau_j$) at two wavelengths ($\lambda_i$, $\lambda_j$) when both are on a log-scale:

$$\mathcal{A}_{i-j} = -\frac{\log(\tau_i/\tau_j)}{\log(\lambda_i/\lambda_j)} \tag{5}$$

### 2.4   Aerosol extinction sensitivity experiments

Following the Eq.4, the extinction coefficient is a product of the three individual terms: $\mathcal{Q}_{\text{ext}}$, $\mathcal{S}$, and $\mathcal{C}$. To explore the sensitivities of the extinction coefficients to aerosol radius, *Mie* calculations are performed for aerosols at various sizes and with different refractive indices. The calculation is based on the off-line *Mie code* proposed by De Rooij and Van der Stap (1984). It is slightly different from the *Mie* code (Boucher, 1998) coupled in our LOTOS-EUROS model where the $\mathcal{Q}_{\text{ext}}$ calculations from the Mie model are stored in lookup tables for higher computational efficiency. Recently, we showed both of the codes

give the exact same result. Aerosol size distributions in LOTOS-EUROS are described using a modal approach. Each mode is represented by a mean geometric radius $r_g$ and a geometric standard deviation $\sigma_g$ as has been illustrated in Table .4. In the following aerosol extinction sensitivity tests, the independent variable $r_g$ is varied over the range from 10 nm to 4 $\mu$m with a step of 10 nm. The control variables associated with the conversion of mass concentration to AOD can be found in Table 2.

The radius-dependent variation in the extinction efficiency $\mathcal{Q}_{ext}$, total cross section per unit mass $\mathcal{S}$, extinction coefficients

$\epsilon_{\text{ext}}$, and the Ångström exponent $\mathcal{A}$ of the five aerosol species for two relative humdities (RH=0/0.8) and five wavelengths ($\lambda =$ 440, 470, 550, 650 and 870 nm) are calculated and presented in detail in Fig. S1 to Fig. S5, respectively. These five incident wavelengths are employed for the aerosol optical property retrieval either in AERONET or in MODIS data collection. Parts of the representative results are also shown in Fig. 1 to illustrate the relationship between aerosol extinction and radius.





### 2.4.1 Extinction efficiency

The relationship between the extinction efficiency $\mathcal{Q}_{ext}$ and mean geometric radius for sulfate and organic carbon aerosols are presented in Fig. 1(a). Small particles are very poor scatters and also their light absorption capacity is well below what we may expected from their physical size; the extinction efficiency of particles much smaller than the wavelength of light is therefore

$\ll 1$. Very large particles remove twice as much light as we may expect from their geometrical cross section. Physically, this is explained by the sum of light scattered by reflection and refraction within the particle plus the diffracted component that is lost from the direct beam. Particles are most active, or have the highest extinction efficiency, when their sizes are in the range of the wavelength. For single particles that do not absorb light the scattering may be four times more efficient than their real size suggests. When considering distributions of particles as in Fig. 1 than the combination with less optically efficient particles

reduces the maximum in the extinction efficiency. A particle size distribution also removes so well-known maxima, minima, and secondary order ripples in the extinction efficiency curve that are due to interference of light with spheres that have just the right dimension to diffract and transmit light.

For absorbing aerosol the peak in the scattering efficiency is reduced (e.g., Hansen and Travis (1974)). For non-absorbing aerosol the scattering efficiency curve seems to move to the left for higher real parts of the refractive index, see e.g. Figure

2 in Moosmüller and Ogren (2017) and our Fig. 1a where the extinction efficiency curve of oc (mreal = 1.53, $\lambda$ = 550 nm) peaks at a smaller size distribution than the extinction efficiency curve of so4 (mreal = 1.43, $\lambda$ = 550 nm). Next to the chemical constituent, the extinction efficiency also depends on the ratio of particle size and wavelength of light, which are usually expressed as a size parameter $x = 2\pi \cdot r_g/\lambda$. As can be seen in Fig. 1a, although the peak of the extinction efficiency is found at a larger size distribution when the incident wavelength is changed from 550 nm to 870 nm, the peak is actually found at the

fixed size parameter $x$ as for the same aerosol species.

Also in Fig. 1a we show the effect of a higher humidity (RH=0.8). The hydrophilic nature of so4 leads to water uptake and thus physical size growth. The smaller dry particles, distributions left of the extinction efficiency peak, thus grow into the more optically active region, i.e. the curve and peak move to the left. On the other hand, the uptake of water, with a real part of the refractive index of 1.33, will make the curve move to the right. The net effect of water uptake is a move to the left. Hence the

change in size is more important than the change in refractive index.

### 2.4.2 Total cross section

Panel b in Fig. 1 plots the total cross section per dry mass $\mathcal{S}$ for sulfate, organic carbon, and black carbon aerosols at different size distributions. The total cross section $\mathcal{S}$ is a product of the mean cross section and total number per dry mass. The former is proportional to the square of the geometric mean radius $r_g$ while the later is proportional to the negative cubic power of $r_g$.

In terms of $\mathcal{S}$, aerosols at a larger size distribution are less effective in diminishing the total solar radiation compared to finer aerosol bins. A steady decline in $\mathcal{S}$ is therefore found with an increase of the aerosol size distribution over all the species.

For the inorganic aerosols, the hydrophilic characteristic tends to efficiently increase $\mathcal{S}$. Take the so4 (hygroscopicity index $\mathcal{K}$ = 0.8) for instance, there is a 1.61 growth in the diameter when they are surrounded with a wet atmosphere (RH=0.8)




following the aerosol diameter hygroscopic growth function $f(\mathrm{RH}) = \{1 + \mathcal{K} \cdot \mathrm{RH}/(1-\mathrm{RH})\}^{1/3}$ (Petters and Kreidenweis, 2007), which is equally a 2.60 times growth in $\mathcal{S}$ here. The total cross section $\mathcal{S}$ of oc is less sensitive to relative humidity since it has a much lower hygroscopicity index ($\mathcal{K} = 0.1$). For hydrophobic aerosols like dust and black carbon, their total cross section won't change when they are moved to a wet atmosphere.

We also show the effect of mass density on the $\mathcal{S}$ calculation. Physically, for aerosols with a fixed mass, the ones have lower mass density are at larger size distribution and hence results higher total cross section. The mass density of sulfate, organic carbon, and black carbon aerosols are 1841, 2000 and 2000 kg/m$^3$. In terms of $\mathcal{S}$, sulfate aerosol bins are slightly more efficient in diminishing the total solar radiation than the organic carbon and black carbon aerosols when they are at the same size distribution.

### 2.4.3   Extinction coefficient

The extinction coefficients $\epsilon_{\mathrm{ext}}$ are then calculated as a product of $\mathcal{Q}_{ext}$, $\mathcal{S}$ and a given dry aerosol concentration $\mathcal{C}$ (1 g/m$^3$) following Eq. 4. The relationship between the aerosol extinction coefficients $\epsilon_{ext}$ and the size distribution $r_g$ are shown in Fig. 1(c). In general, $\epsilon_{\mathrm{ext}}$ presents an up-and-down pattern: aerosols at a small size distribution result in a low $\epsilon_{\mathrm{ext}}$ due to their inactive extinction efficiency $\mathcal{Q}_{ext}$ (see panel a); while particles at a large size distribution that leads to a small total cross section $\mathcal{S}$

would result in a low $\epsilon_{\mathrm{ext}}$ as well (see panel b). Particles with geometric mean radius ranging from tens to hundreds nanometers are most active in diminishing the solar radiation. The effect of chemical constituent on the extinction coefficient can also be found in in panel c. Black carbon reaches a higher peak at a smaller size distribution than organic carbon and inorganic carbon due to its high extinction efficiency $\mathcal{Q}$ curves at the small size. Water absorption (RH=0.8) makes the hygroscopic aerosols become more efficient in diminishing the light absorption and scattering, e.g., the $\epsilon_{\mathrm{ext}}$ curve and peak of so4 aerosol moves to

the upper left.

    The efficient coefficients vary significantly at different sizes distribution. For instance, $\epsilon_{\mathrm{ext}}$ of so4 at incident wavelength of $\lambda = 550$ nm (RH = 80%) reaches 12.21 (1/m) at a geometric mean radius of $r_g = 110$ nm; it reduces rapidly to 4.73 (1/m) at $r_g$ = 350 nm. Therefore, the 3D conversion from mass concentration to AOD in aerosol models is highly sensitive to the aerosol size distribution. For a fair comparison between the model AODs and actual AOD observations, the aerosol size should remain

consistent in the AOD operator and the satellite AOD retrieval algorithm.

### 2.4.4   Ångström exponent

Following Eq.5, the Ångström exponent is calculated using the extinction coefficients at two incident wavelengths. Ångström exponent at 870-440 nm and 650-470 nm are calculated in this study, the former is used in the AERONET Ångström product, while the latter corresponds to the MODIS Ångström observational wavelengths. The size-dependent Ångström curves for

inorganic (so4), organic carbon, and black carbon aerosols are presented in panel d in Fig. 1. It is worth noting that a continual decline is observed in the Ångström exponent curve for all the species when the aerosol size distribution grows up to 400 nm. Subsequently, the Ångström curve remains stable with an increase in the aerosol geometric mean radius. It is evident that the Ångström data contains valuable aerosol size information. Apart from the coarse-mode dominated dust and sea salt aerosols,





inorganic, black carbon, and organic carbon aerosols are believed to be fine-mode dominated which exhibit a mean radius of less than 400 nm. Therefore, Ångström is a key quantitative indicator of aerosol bin sizes.

In real situations, airborne aerosols are a mixture of several species. The extinction coefficients of mixed aerosols equal the sum of $\epsilon_{ext}$ for all species. The Ångström exponent is interpolated using the integrated $\epsilon_{ext}$ at two different incident bands, thus indicating the size distribution of mixed aerosols.

## 3 Measurements

Measurement data from three different sources are used in this study. Ångström exponent and AOD measurements from MODIS Deep Blue products have been used in the assimilation system that will be described in section 5. The corrections made to the model aerosol size information are validated through a comparison with the AERONET optical products, while the corrections made to the surface aerosol mass concentration simulation are evaluated using independent ground PM$_{2.5}$ observations. The data sets and relevant information are summarized in Table 3.

### 3.1 MODIS

In this study, Deep Blue aerosol products (Hsu et al., 2013; Sayer et al., 2014) of the Moderate Resolution Imaging Spectro-radiometer (MODIS) C6.1 data suite have been used in the aerosol assimilation system. Its Ångström exponent $\mathcal{A}_{650-470}$ is assimilated to estimate the aerosol radius, while its AOD data $\tau_{550}$ is assimilated to estimate the aerosol mass concentration field. The assimilation of MODIS aerosol properties is conducted in the original MODIS observational space. Specifically, each of the MODIS measurements is compared to the the aerosol simulation at the grid cell that holding the MODIS pixel.

Snapshots of MODIS AOD and Ångström captured on July 24 (between 10:00-11:00) are presented in Fig. 2(a)-(b). Most MODIS AOD observations stay in the range from 0.05 to 0.8. Ångström exponents exhibit less spatial variability, and most values stay around 1.2 to 1.6. However, there are still some low Ångström exponents; for instance, the ones in the green colored region in Fig. 2(b), which have been validated as inconsistent measurements in Section 3.2.

### 3.2 AERONET

The MODIS product is first evaluated through a comparison with ground-based observations collected from the Aerosol Robotic Network (AERONET). At the sites of this network, total number of AOD columns are measured using ground sun-photometers. AOD and Ångström measurements at Cabauw and Leipzig have been used for validation. The location of these two AERONET sites are marked in Fig. 5. Figure 3 and Figure 4 show the time series of AERONET AOD and Ångström observations at these two sites from the Level 1.5 product (cloud-screened and quality controlled). MODIS AODs and Ångströms at 500 nm are also shown as an average over the $0.1° \times 0.1°$ model grid cell in which the site is located.

The time series reveal that the MODIS AOD and Ångström observations match the spread in the AERONET observations well. There are also some inconsistent MODIS Ångström values that did not match the AERONET measurements all the time. Specifically, we have several MODIS Ångström measurements around 0 at the Leipzip site, as shown in the green colored



**Figure 1.** Aerosol extinction vs. mean geometric dry radius $r_g$. (a): extinction efficiency $Q_{ext}$; (b): total cross section per unite dry mass $S$; (c): extinction coefficients $\epsilon_{ext}$; (d): Ångström exponent $A$. Note that $\lambda$ represents the incident wavelength. Abbreviations (bc: black carbon; so4: sulfate aerosols; oc: organic carbon; RH: relative humidity)





**Figure 2.** MODIS AOD and Ångström, Control LOTOS-EUROS $AOD_{550}$ and $Ångström_{650-470}$. Although LE AOD and Ångström can be simulated anywhere, they are only projected into the MODIS space for making the comparison easier. Gray pixels indicate observation vacancy.





region in Fig. 4(b); meanwhile, the nearby MODIS Ångström exponents and AERONET measurements exhibit high levels. The 0 MODIS Ångström exponent at Leipzig on July 24 refers to the pixels in the green colored region (Fig. 2(b)). These local inconsistent Ångström observations are supposed to occur due to the retrieval error, which might prevent us from exploring fine-scale aerosol size distributions in Section 5.1.2.

5      AERONET AOD and Ångström observations are interpolated at wavelengths different from those used to interpolate the MODIS aerosol product; the details are presented in Table. 3.

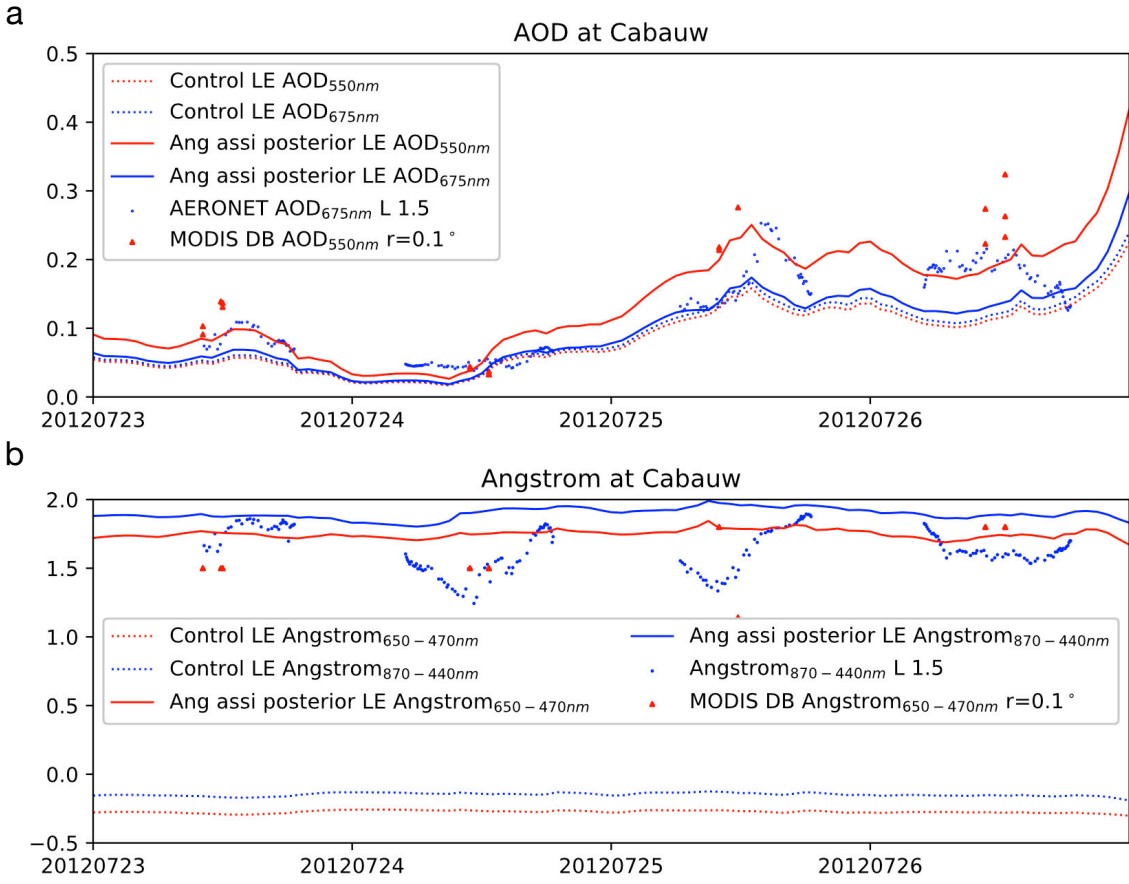

**Figure 3.** AERONET, MODIS and LOTOS-EUROS prior AOD (a) and Ångström (b) at Cabauw ($51°58'$ N, $04°55'$ E). Note that $r$ defines the radius for mapping the MODIS product into test sites. Observations from AERONET and the consistent LE simulations are marked in blue; observations from MODIS and the comparable simulations are shown in red.

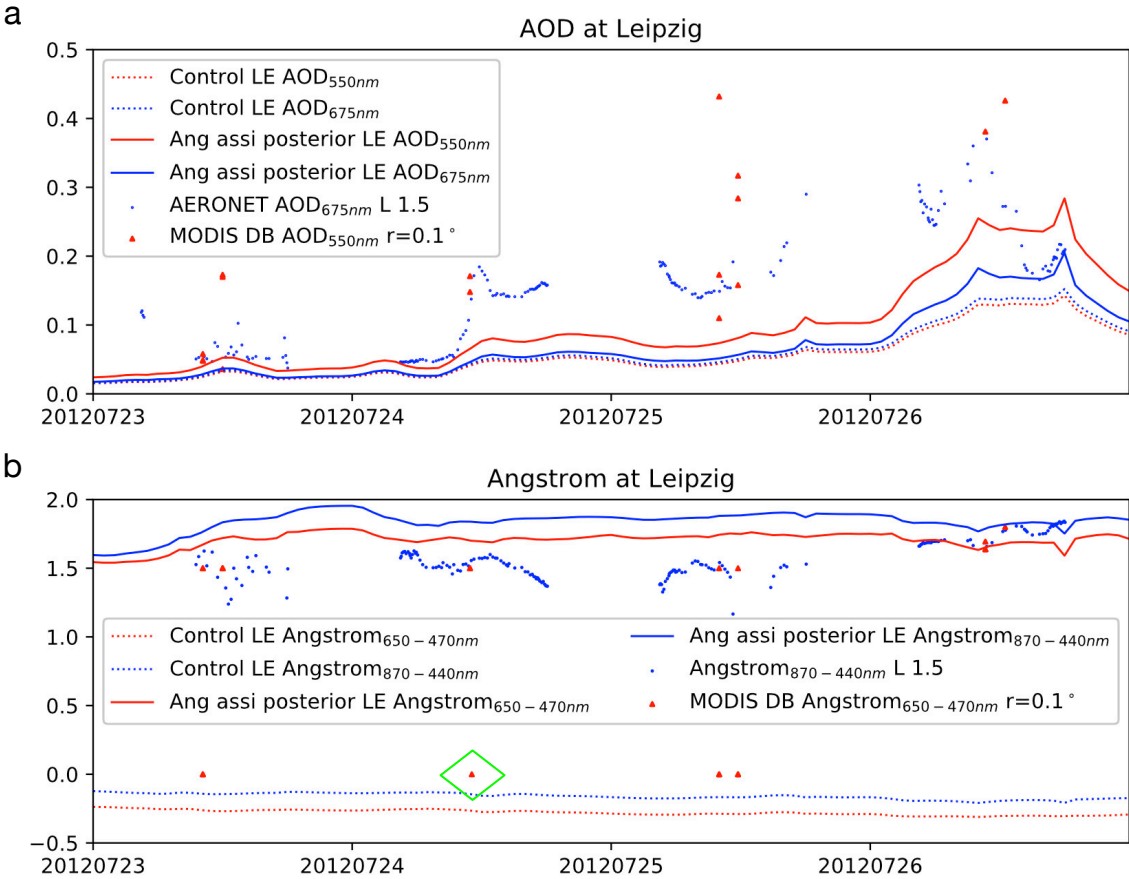

**Figure 4.** AERONET, MODIS and LOTOS-EUROS prior AOD (a) and Ångström (b) at Leipzig ($51°21'$ N, $12°26'$ E). Note that $r$ defines the radius for mapping the MODIS product into the test site.

### 3.3 Ground PM$_{2.5}$ concentration

The focus of aerosol models and remote sensing data assimilation is to achieve accurate estimation of the aerosol state field, which would subsequently forward an accurate forecasting of aerosol mass concentrations. In this study, hourly PM$_{2.5}$ observations over 151 EU air quality ground stations have been collected to validate the model simulations on surface aerosol concentrations. The distributions of these ground stations are shown in Fig. 5. Although other air quality monitoring stations were available during our study, they only provide daily-averaged aerosol measurements. They have large uncertainties for representing the instant aerosol loading measured by the MODIS instruments, and therefore is not used here.

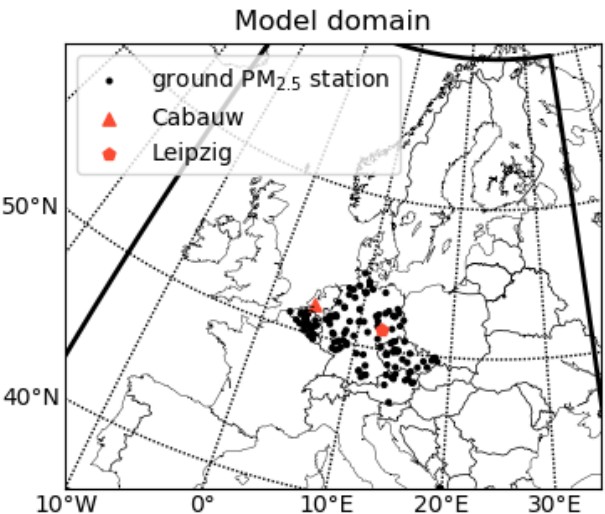

**Figure 5.** Model domain and locations of ground PM$_{2.5}$ stations and AERONET sites at Cabuaw and Leipzig.

**Table 3.** Measurements for assimilation and validation

| Measurement | descriptions |
|---|---|
| MODIS AOD | Deep Blue over land product, $\lambda = 550$ nm |
| MODIS Ångström | Deep Blue over land product, $\lambda_1 = 650$ nm; $\lambda_2 = 470$ nm |
| AERONET AOD | Level 1.5, $\lambda = 675$ nm |
| AERONET Ångström | Level 1.5, $\lambda_1 = 870$ nm; $\lambda_2 = 440$ nm |
| ground PM$_{2.5}$ | hourly |

# 4 Model description

The satellite observations of AOD's and derived Ångström exponents will be assimilated with model simulations to obtain the best possible representation of aerosol concentrations. For the simulations a regional chemical transport model (CTM) will be used that is described in section 4.1. As reference for the assimilation experiments, a standard simulation has been performed that is described in section 4.2.

## 4.1 Model description and aerosol size distribution

The regional chemistry transport model (CTM) LOTOS-EUROS has will be used to simulate aerosol concentrations. This simulation model has been used for a wide range of applications related to air quality simulations, forecasts and scenario studies, both inside and outside Europe (Manders et al., 2017).





**Table 4.** Geometric means and standard deviations used for aerosol radius size distributions in this study. The columns for (apriori) uncertainty and posterior geometric mean are used as input and obtained as output of the assimilation procedure.

| Species | mode | geom. stdv. | geom. mean radius | | |
| --- | --- | --- | --- | --- | --- |
| | | | prior | uncertainty | posterior |
| | | [$\mu$m] | [$\mu$m] | [$\mu$m] | [$\mu$m] |
| inorganic ($SO_4$, $NO_3$, $NH_4$) | fine | 1.59 | 0.350 | 0.070 | 0.0904 |
| inorganic ($SO_4$, $NO_3$) | coarse | 2.00 | 2.500 | 0.100 | 2.713 |
| black carbon | fine | 1.59 | 0.350 | 0.070 | 0.1105 |
| black carbon | coarse | 2.00 | 2.500 | 0.100 | 2.712 |
| organic. primary | fine | 1.59 | 0.350 | 0.070 | 0.0708 |
| organic, primary | coarse | 2.00 | 2.500 | 0.100 | 2.539 |
| dust, sea salt | fine 1 | 1.59 | 0.165 | | |
| dust, sea salt | fine 2 | 1.59 | 0.350 | | |
| dust, sea salt | coarse 1 | 2.00 | 1.500 | | |
| dust, sea salt | coarse 2 | 2.00 | 2.500 | | |
| dust, sea salt | coarse 3 | 2.00 | 4.000 | | |

For the current study, the LOTOS-EUROS model has been configured over a domain from 35°N to 70°N and 15°W to 35°E (shown in Fig. 5), with a resolution of 0.25°× 0.25° (about 15 × 25 km at these latitudes). In the vertical a simple mixing layer approach is used with in total only 5 layers: a surface layer of 25 m, a mixing layer, two reservoir layers, and a top layer that reaches to an altitude of 5 km. This rather coarse configuration allows fast simulation of the main trace gas and aerosol

concentrations. Physical processes included are emission, advection, diffusion, dry and wet deposition, and sedimentation. Anthropogenic emissions of trace gases and aerosols are taken from a TNO emission inventory (Kuenen et al., 2014). The partitioning of nitrate and ammonium between the gas and aerosol phase is described using ISORROPIA (Fountoukis and Nenes, 2007). Natural emissions of dust and sea salt aerosols are calculated online given surface characteristics and meteorology.

The gas-phase chemistry is based on a carbon-bond mechanism. Aerosol concentrations are represented by 21 different

species corresponding to particles with a specific chemical composition within a certain size mode. A size mode characterizes a distribution of aerosol radii following a log-normal distribution, defined by its geometric mean and standard deviation. The prior geometric mean and standard deviation of the aerosol species are provided in Table. 4. For most aerosol types two size modes are defined to characterize a fine and a coarse model; for dust and sea-salt aerosols, two fine and three coarse modes are used that allow more detailed modelling of emission and deposition processes. The table also includes numbers for uncertainty

and posterior estimates of the geometric mean, which are input and output of the assimilation procedure.

Our *hybrid* AOD assimilation method that will be described in section 5 has been tested over Europe for the period July 23 to 26 in 2012. This period was chosen because hardly any clouded pixels were present in these four days, and therefore many high quality satellite observations were available.





## 4.2  Prior simulation

The aerosol concentrations during the evaluation period have been simulated with the the LOTOS-EUROS model using a standard configuration. This simulation will serve as the *prior* simulation for the assimilation, also referred to as the *background* or *control* simulation. For the aerosol size distributions, the geometric mean radii from the *prior* column in Table 4 are used.

A snapshot of the AOD and Ångström exponent simulations for a single hour are shown in Fig. 2(c)-(d). To allow better comparison with the corresponding MODIS observations, the model simulations are only shown where observations are available. Compared to the MODIS observations, the simulation shows a strong underestimation of AOD. Most of the model simulated AOD values are less than 0.2, while observations reach values up to 0.8. Also the simulated Ångström exponent strongly under estimates the MODIS observations. Almost all simulated Ångström exponents are smaller than zero, while the observations
are in a range of of 1.2 to 1.6.

Since AOD scales linear with the aerosol concentrations, the under estimation of the observed AOD suggests a lack of aerosol in the model. There are many uncertainties in the model that could explain a too low aerosol load: absence of secondary organic aerosols, under estimation of emissions, or too strong deposition and sedimentation. However, also the simulation of AOD from the concentrations is uncertain, for example because it relies on the assumed size distributions. The underestimation
of the Ångström exponents by the model hints on this too: according to the relationship between Ångström and radius shown in Fig. 1(d)), this under estimation might be a result of assuming too large aerosol sizes in the *prior* model. This study will focus on the later issue first when trying to improve the simulations of AOD; when optimal size distributions are found, the next step is to adjust the emissions in order to change the aerosol load.

## 5  Ångström and AOD assimilation

The observations of AOD and Ångström exponent have been assimilated with the model simulations in order to obtain improved aerosol concentrations fields. The assimilation procedure is described first, followed by the results of the assimilation experiments.

### 5.1  Assimilation setup

The data flow in the assimilation experiments is illustrated in Fig. 6. Two different assimilation configurations are distin-
guished: *Standard AOD assimilation* that takes only AOD observations into account, and the new *hybrid AOD assimilation* that also includes Ångström exponent observations. In both systems, the observations are used to obtain *posterior* aerosol mass concentrations that should better represent the actual concentrations. These could be used used as initial conditions for a *forecast*, a simulation with a standard model configuration starting from the *posterior* concentrations; in this study, no *forecast* experiments are performed however. The assimilation of AOD observations is done in the same way in both systems; however,
in the *hybrid* system, the assumed aerosol size distributions for the AOD simulations are obtained through an extra *Ångström analysis*. To distinguish the AOD assimilations from each other, the AOD assimilation using the AOD operator based on prior





aerosol size distribution is refered to as the *AOD analysis*, while the AOD assimilation based on the posterior aerosol size distribution from the *Ångström analysis* is refered to as the *Hybrid analysis*.

Both the AOD and hybrid assimilation methodology will been applied to aerosol simulations over Europe from July 23 to 26 in 2012. A single assimilation window of 4 days is used, collecting all available observations during the period, and optimizing 5 AOD or radii and AOD once. A systematic study with longer model periods and more assimilation cycles would help to further understand temporal variation of the aerosol radius; this will be a part of our future work.

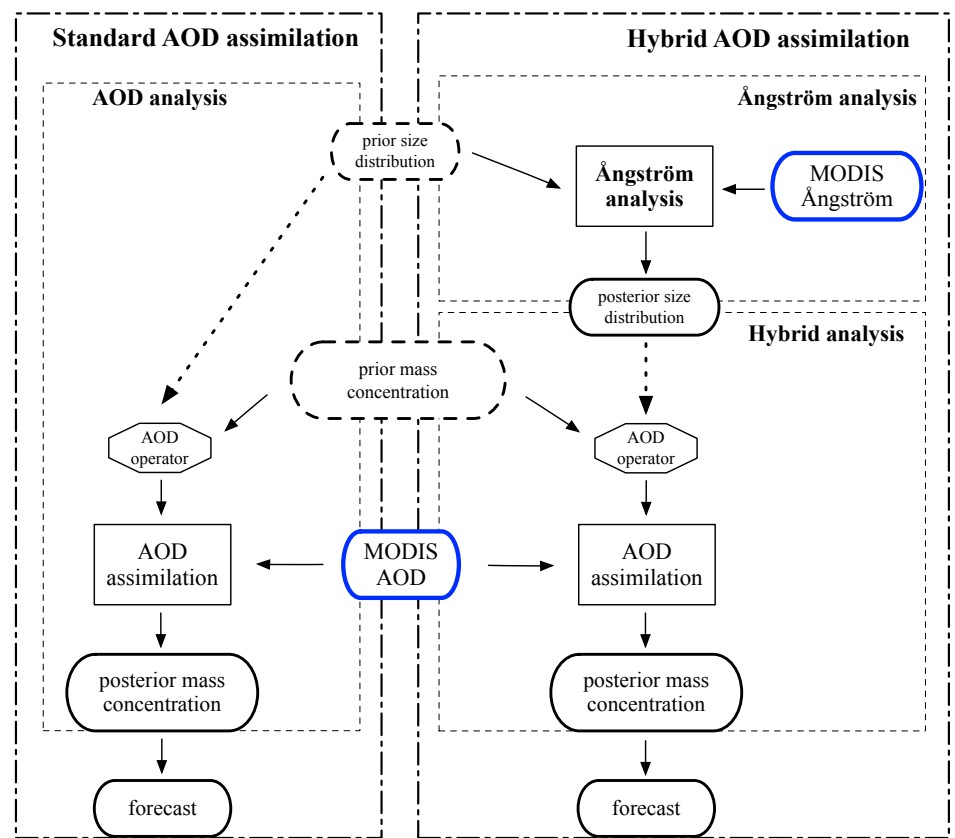

**Figure 6.** *Standard* and *hybrid* AOD data assimilation systems.

### 5.1.1 AOD assimilation methodology

The assimilation of AOD observations is performed using a 3D variational approach (Kalnay, 2002). The goal of the assimilation is to find *posterior* 3D aerosol concentrations $X$ that provide the best simulation of AOD compared to the observations. 10 In the variational approach, the optimal (analyzed) concentrations $X_a$ are defined as an increment $\delta X_a$ on the *prior* (or background) concentrations $X_b$:

$$X_a = X_b + \delta X_a \tag{6}$$





For simplicity (and computational efficiency) it is assumed that in the analyzed concentrations the ratios between different aerosol concentrations remains the same as in the background, and that the vertical profile of the concentrations remains the same too. Under this assumptions it is sufficient to find an optimal 2D AOD field first, and compute from that all 3D mass concentration increments. For this, first the AOD should be simulated from the background concentrations:

$$5 \quad \boldsymbol{\tau}_b^m = \mathcal{H}(\boldsymbol{r}_b)\, \boldsymbol{X}_b \tag{7}$$

In here, $\mathcal{H}$ is the AOD observation operator that depends on the assumed aerosol size distribution parameters, which for the *standard* assimilation are the background or *prior* size distributions collected in $\boldsymbol{r}_b$. In practice, $\mathcal{H}$ represents the AOD simulation from the mass concentration in the model as well as the observation selection operator. The latter is a matrix filled with 0/1 used for selecting the model AOD into the AOD observation space (white pixels excluded). To distinguish these two parts, we directly perform AOD assimilation analysis at pixels where AOD measurements are available. Hence, $\mathcal{H}$ purely represents the conversion of aerosol mass concentration to AOD solely for this study.

The optimal (analyzed) AOD field $\boldsymbol{\tau}_a^m$ is then defined as an increment of the background values:

$$\boldsymbol{\tau}_a^m = \boldsymbol{\tau}_b^m + \delta\boldsymbol{\tau}_a^m \tag{8}$$

When the optimal AOD increment $\delta\boldsymbol{\tau}_a^m$ is found, the optimal concentration increment is calculated from:

$$15 \quad \delta\boldsymbol{X}_a(i,k,s) = \boldsymbol{X}(i,k,s) \cdot \frac{\delta\boldsymbol{\tau}_a^m(i)}{\boldsymbol{\tau}_b^m(i)} \tag{9}$$

where $i$ and $k$ denote the spatial and vertical location in the 3D model fields, and $s$ denotes the aerosol tracer.

The optimal AOD increment $\delta\boldsymbol{\tau}_a^m$ is defined as the field that minimizes the cost function:

$$J(\delta\boldsymbol{\tau}_a^m) = \frac{1}{2}(\,\delta\boldsymbol{\tau}_a^m\,)^{\mathrm{T}}\,\mathbf{B}_{\boldsymbol{\tau}}^{-1}\,(\,\delta\boldsymbol{\tau}_a^m\,) + \frac{1}{2}(\boldsymbol{\tau}_b^m + \delta\boldsymbol{\tau}_a^m - \boldsymbol{\tau})^{\mathrm{T}}\,\mathbf{R}_{\boldsymbol{\tau}}^{-1}\,(\boldsymbol{\tau}_b^m + \delta\boldsymbol{\tau}_a^m - \boldsymbol{\tau}) \tag{10}$$

The first part of the cost function defines a penalty on a perturbation from the background AOD. The background error covariance $\mathbf{B}_{\boldsymbol{\tau}}$ defines the weight of the penalty; how this covariance is defined is described below. The second part of the cost function defines a penalty on a deviation of the simulated AOD from the observations $\boldsymbol{\tau}$; the observation error covariance $\mathbf{R}_{\boldsymbol{\tau}}$ defines the weight of the penalty on an observation-minus-simulation mismatch.

The background and observation error covariance $\mathbf{B}_{\boldsymbol{\tau}}$ and $\mathbf{R}_{\boldsymbol{\tau}}$ together define the optimal solution of the minimization problem of Eq. (10), and are therefore the most important entities of the data assimilation system (Kalnay, 2002). In the background covariance $\mathbf{B}_{\boldsymbol{\tau}}$, the main diagonal defines the assumed variance of the model AOD, while the off-line elements represent the correlations between two AOD values in different grid cells. In this study the focus is on using the available AOD observations to obtain insight in the validity of the assumptions on the aerosol size distribution; correlations between grid cells are therefore simply ignored, and all optimizations are done per grid cell. The background covariance is therefore implemented as a diagonal matrix. We have used 30% to characterize the uncertainty of our background AOD simulation, with a minimum uncertainty 0.2 to prevent the posterior solution from becoming too close to the low-value AOD prior simulation:

$$\mathbf{B}_{\boldsymbol{\tau}}(i,i) = \max(\,0.2,\ 0.3\,\boldsymbol{\tau}_b(i)\,)^2 \tag{11}$$





The observation representation error covariance $\mathbf{R}_\tau$ defines the errors in the observed AODs from instrument and retrieval uncertainties. These errors are assumed to be independent from each other, and therefore $\mathbf{R}_\tau$ is modeled as a diagonal matrix. The diagonal elements are directly taken from the MODIS Deep Blue product.

### 5.1.2 Hybrid assimilation methodology

The *Hybrid assimilation* is carried out by sequentially implementing the *Ångström analysis* and the *AOD analysis*. The *Ångström assimilation* focuses on estimating the aerosol size distribution, and is performed through minimizing the following cost function:

$$J(\boldsymbol{r}_a) = \frac{1}{2}\left(\boldsymbol{r}_a - \boldsymbol{r}_b\right)^{\mathrm{T}} \mathbf{B}_{\boldsymbol{r}}^{-1}\left(\boldsymbol{r}_a - \boldsymbol{r}_b\right) + \frac{1}{2}\left(\mathcal{M}\left(\boldsymbol{r}_a\right)\boldsymbol{X}_b - \boldsymbol{\mathcal{A}}\right)^{\mathrm{T}}\mathbf{R}_{\mathcal{A}}^{-1}\left(\mathcal{M}\left(\boldsymbol{r}_a\right)\boldsymbol{X}_b - \boldsymbol{\mathcal{A}}\right) \tag{12}$$

The first part defines the mismatch between the optimal aerosol radius and the prior size assumptions, while the second part

quantifies the penalty from the MODIS Ångström observations.

     Vectors $\boldsymbol{r}_a$ and $\boldsymbol{r}_b$ denote the analyzed and prior aerosol radius over the 21 aerosol bins in Table 4. The aerosol radii are assumed to be spatially and temporally constant during the short period used for the experiments. Spatially varying radii would of course allow the assimilated Ångström exponent to better fit the MODIS Ångström exponent. However, the locally inconsistent MODIS Ångström observations found during comparison with AERONET observations in Section 3.2 would

introduce strong local mis-adjustments in case a (large) spatial degree of freedom is allowed. Introducing spatial variations also requires information on spatial correlations and will increase computational costs; hence, this aspect has not been explored in this study. The radii of the different aerosol species are also assumed to be independent on each other. The background covariance matrix $\mathbf{B}_{\boldsymbol{r}}$ is therefor diagonal, with elements set to the the square of the uncertainties listed in Table.4. The uncertainties are chosen empirically and capable of resolving the the mismatch between the observed and simulated Ångström

exponents. The MODIS Deep Blue product used in this study only provides aerosol measurements over lands, and dust storms are sure to be absent in our evaluation period. Our aerosol simulating results is less effected by the dust and sea-salt. Therefore, both the dust and sea-salt aerosol bins are assumed certain and not estimated in this study.

     In the second part of the cost function Eq. (12), the operator $\mathcal{M}\left(\boldsymbol{r}_a\right)$ represents the Ångström simulation from the model state $\boldsymbol{X}_b$, which depends on the aerosol size distribution $\boldsymbol{r}_a$. Covariance matrix $\mathbf{R}_{\mathcal{A}}$ defines the weight of the penalty for a

mismatch between the simulation and the Ångström observation $\mathcal{A}$. Similar as for the AOD observation representation error it is defined as a diagonal matrix under the assumption that all Ångström measurements are independent from each other. The diagonal elements are set to the square of 0.3, which is an empirical chosen value obtained from a comparison between the MODIS Ångström and AERONET Ångström measurements.

     The aerosol radius $\boldsymbol{r}_a$ that minimizes the cost function Eq. (12) is obtained using a 4DvEnvar method, an updated model

AOD simulation is then obtained via:

$$\boldsymbol{\tau}_h^m = \mathcal{H}(\boldsymbol{r}_a)\,\boldsymbol{X} \tag{13}$$





Following the same procedure as for the standard *AOD assimilation*, a new AOD analysis increment is obtained by minimization of a cost function similar to Eq. (10):

$$J(\delta\boldsymbol{\tau}_h^m) \;=\; \frac{1}{2}\,(\delta\boldsymbol{\tau}_h^m)^{\mathrm{T}}\;\mathbf{B}_{\boldsymbol{\tau}}^{-1}\;(\delta\boldsymbol{\tau}_h^m) \;+\; \frac{1}{2}\,(\boldsymbol{\tau}_h^m \;+\; \delta\boldsymbol{\tau}_h^m \;-\; \boldsymbol{\tau})^{\mathrm{T}}\mathbf{R}_{\boldsymbol{\tau}}^{-1}(\boldsymbol{\tau}_h^m \;+\; \delta\boldsymbol{\tau}_h^m \;-\; \boldsymbol{\tau}) \tag{14}$$

With the optimal increments of AOD simulation $\delta\boldsymbol{\tau}^m$ obtained through minimizing the above penal function 14, The increments of aerosol mass concentration $\boldsymbol{X}$ can be calculated via:

$$\delta\boldsymbol{X}_h(i,k,s) = \boldsymbol{X}(i,k,s)\cdot\frac{\delta\boldsymbol{\tau}_h^m(i)}{\boldsymbol{\tau}_h^m(i)} \tag{15}$$

## 5.2 Ångström analysis results

The *Ångström analysis* is performed first following the previously described methodology. The posterior radii of the 21 aerosol species are listed in Table 4. Compared to the prior assumptions, the Ångström analysis estimates smaller radii for the fine modes. The initial assumption of a geometric mean radius of 350 nm is reduced to about 90 nm for the inorganic aerosol in the fine mode, to 110 nm for black carbon, and 71 nm for organic and primary aerosols. The sizes assumed for the coarse mode are slightly increased from 2.5 $\mu$m initially to 2.7 $\mu$m for inorganic and black carbon aerosols, and remain about the same for the organic and primary aerosols.

Figure 7 shows posterior simulations of AOD and Ångström exponent using the optimized radii from the *Ångström analysis*. The simulated posterior Ångström exponents are now in the same range as the MODIS observations in Fig. 2(b). Most of the values are in the range of 1.2 to 1.6, while the prior simulation produced negative values. The simulated Ångström exponents field is rather smooth since no spatial variation in the analysis was allowed. The AOD values simulated using the optimized radii (Fig. 7(a)) have increased compared to the prior simulation (Fig. 2(c)). The simulated AOD is however still under estimating the MODIS observations (Fig. 2(a)).

Time series of the posterior AOD and Ångström exponent simulations in the two AERONET sites Cabauw and Leipzig are included in Fig. 3 and Fig. 4, respectively. Compared to the prior simulation, the posterior Ångström simulations are much closer to the independent AERONET observations. However, the temporal variability that is seen in the AERONET observations (blue dots) is not reproduced by the model. This could be explained by the use of aerosol radii that are constant in time. A temporal varying aerosol size is for the current application not feasible since it is based on MODIS data which has only a limited number of overpasses per day. The simulated AOD in the two sites is increased when using the posterior radii, but still an under estimation is present.

## 5.3 AOD analysis results

Two assimilations of MODIS AOD have been performed using either the prior aerosol size distribution (*AOD analysis*) or using the posterior aerosol radius determined from the *Ångström analysis* (*hybrid analysis*). Figure 8(a)-(b) shows examples of the posterior AOD simulations. Although the analyses are based on different prior AOD fields (see Figure 2(c) and Figure 7(a)), the posterior AOD values are very similar since they are optimized to represent the same MODIS AOD observations.





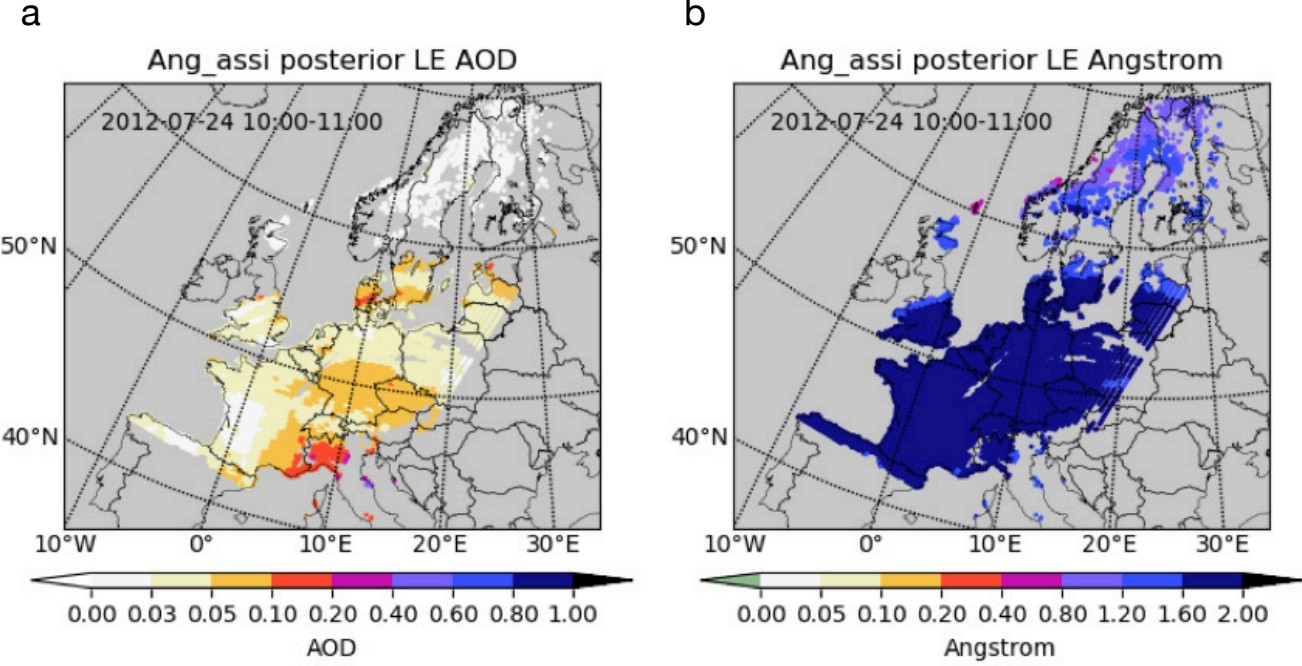

**Figure 7.** Posterior LOTOS-EUROS simulations of AOD (a) and Ångström exponent (b) at July 24, 10:00 to 11:00 after *Ångström analysis*.

However, the increments that were applied to the aerosol mass concentrations could be very diffent. The lower panels of Figure 8 show the increment in surface $PM_{2.5}$ for the same hour as the AOD simulations in the upper panels. While in the *AOD assimilation* the increments range from 12 to 24 $\mu g/m^3$, in increments in *hybrid assimilation* are much lower and range from about 6 to 12 $\mu g/m^3$. This shows that AOD operator (or in this study, the assumed aerosol radii) strongly influence the aerosol

5   mass concentration estimation during assimilations of AOD.

To evaluate the effect of the *AOD* and *hybrid assimilation* on aerosol concentrations, the posterior surface $PM_{2.5}$ simulations are compared to the ground $PM_{2.5}$ observations. Figure 9 shows maps of surface $PM_{2.5}$ measurements, and simulation from the *control* run, the *AOD assimilation*, and the *hybrid assimilation* on July 24 (10:00-11:00). The $PM_{2.5}$ concentrations (a) are underestimated in the control run (b), but strongly increased by the assimilations (c and d). If only AOD is assimilated (c), the

10  surface $PM_{2.5}$ concentrations actually exceed the observations. However, if aerosol radii are optimized first using the *Ångström analysis* in the *hybrid assimilation*, the simulated concentrations are good agreement with the observations. This indicates that although a standard *AOD assimilation* might be able to improve the AOD fields, it does not ensure the improvement on the aerosol mass concentration due to uncertainties in the AOD operator. The *hybrid assimilation* seems better able to relate AOD with aerosol masses since it uses aerosol sizes that are in better agreement with the retrieved Ångström exponent. It would be







**Figure 8.** Posterior AOD simulations (top) and increments of surface PM$_{2.5}$ concentrations (bottom) in ether *AOD assimilation* (left) and *hybrid assimilation* (right) on July 24, 10:00 to 11:00.



**Figure 9.** PM$_{2.5}$ observation (a), prior (b), *AOD analysis* (c) and *Hybrid analysis* (d) surface PM$_{2.5}$ simulation posterior on July 24 2012, 10:00 to 11:00





interesting to see whether the estimated radii are also in better agreement with the assumptions made in the aerosol retrieval algorithm.

To further evaluate the impact of the *Ångström analysis* on PM$_{2.5}$ concentrations, the root mean square error (RMSEs) between observations and simulations has been calculated for each of the hours where MODIS observations were available

in the 4-day window. The result is shown in Figure 10, which also shows the number of MODIS observations available for analysis at each moment. For the *control* simulations the RMSE values are on average about 9 $\mu$g/m$^3$. Assimilation of just the AOD observations actually increases the RMSE at most times, with an average value around 10.8 $\mu$g/m$^3$; this indicates that the relation between aerosol mass and AOD is uncertain here. However, the *hybrid* assimilation is able to decrease the RMSE for almost all occasions to about 8.0$\mu$g/m$^3$.

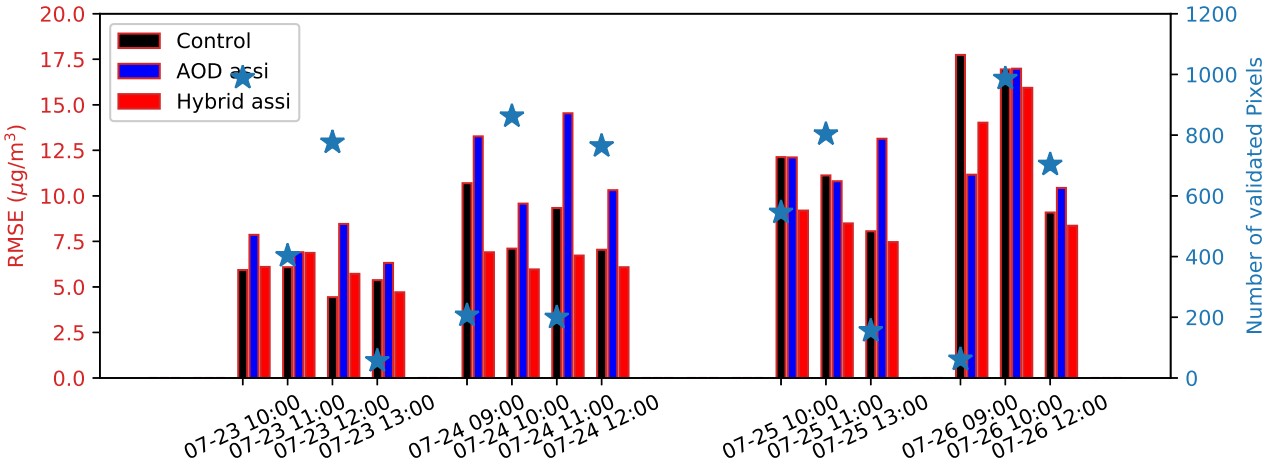

**Figure 10.** RMSE of control, AOD assi and hybrid assi posterior surface PM$_{2.5}$ level vs. the ground PM$_{2.5}$ measurements

## 6 Conclusions

In this study, we investigated the role of aerosol size distribution in calculation of aerosol optical depth from aerosol mass concentrations in simulation model. The assumptions on aerosol size distribution in the model should be in agreement with the assumptions made by the algorithms that retrieve AOD from remote sensing instruments. This is especially essential when remote sensing products such as AOD are assimilated in the model in order improve aerosol mass concentrations.

Aerosol extinction sensitivity tests based on an offline *Mie* code have been performed to test the relation between the aerosol size distribution and the extinction coefficients for different aerosol species (e.g., organic and inorganic, black carbon, mineral dust, and sea salt aerosols), and for incident wavelengths. The results illustrated the high dependence of extinction coefficients on aerosol radius. However, the results also show that the Ångström exponents that is computed based on AOD at two different





wavelengths is a suitable quantitative indicator of aerosol size. The Ångström exponents could therefore be used to improve the assumptions on aerosol size made in the simulation model.

To bring the retrieval and model AOD calculation in better agreement with each other using Ångström expoment, a *hybrid assimilation* methodology is proposed. Different from a *standard AOD assimilation* that directly assimilates AOD observations

and ignores the potential mismatch of the particle radius distribution, the *hybrid* approach first estimates optimal aerosol size parameters by assimilating Ångström exponent observations, before performing the AOD analysis. In both the *AOD* and *hybrid assimilation*, the relative change in AOD obtained from the assimilation is used to scale aerosol mass concentrations. The proposed *hybrid assimilation* has been evaluated by assimilating MODIS Deep Blue AOD in a regional CTM over Europe during a 4-day assimilation window, in the *hybrid assimilation* preceded by an assimilation of corresponding Ångström exponents.

For the *Ångström analysis* that is part of the *hybrid* approach, validation with Ångström exponents retrieved from remote sensing observations from ground based stations from the AERONET network showed strong improvement in simulated Ångström exponents. Since not spatial and temporal variation in aerosol radii was allowed in the experiments, fin scale spatial and temporal variations could not be resolved yet. However, the first order estimate of suitable aerosol radii is shown to be a useful improvement already, and helps to avoid that fine scale inconsistencies in retrieved Ångström expoments have a too

strong impact on results. It is advised that data selection procedures are applied on Ångström exponent observations before these are used in an assimilation.

Assimilations of AOD have been performed without and with a preceding analysis of Ångström exponents to optimize assumed aerosol radii. Both assimilations provide similar posterior AOD simulations since the same MODIS AOD observations are used; however, the assimilations provide different aerosol mass concentration. The posterior surface aerosol mass concen-

tration have been validated by comparison with ground based $PM_{2.5}$ measurements. In our 4-day test, the average RMSE of the simulations in the control run is about 9.3 $\mu$g/m$^3$; when assimilating only AOD this actually increases to 10.8 $\mu$g/m$^3$, which shows that better AOD simulation not necessary implies better aerosol mass representation. If in the *hybrid assimilation* also Ångström exponents are assimilated, the average RMSE decreases to 8.0 $\mu$g/m$^3$.

The experiments show that for assimilation of AOD observations with the goal of improving aerosol mass concentrations it

is essential to take AOD at more than one wavelength into account, for example in the form of Ångström exponents. In this way it is possible to optimize aerosol parameters where the AOD simulations are sensitive to, such as the aerosol size that was the focus of this study. Further optimization of this and other parameters, including spatial and temporal variability, will be subject of future studies. When this has lead to a sufficiently accurate simulation of aerosol optical properties, it will be better possible to optimize aerosol mass concentrations including their vertical profile, and over different aerosol types.

**Data availability**

The $PM_{2.5}$ measurements are from the European Environmental Agency air quality database and accessible via https://www.eea.europa.eu/data-and-maps/data/aqereporting-8. The ground-based AERONET aerosol products are from the AErosol RObotic NETwork and are available at https://aeronet.gsfc.nasa.gov/. The MODIS Deep Blue C6 data suites are available at





https://ladsweb.modaps.eosdis.nasa.gov/. The datasets including measurements and model simulations can be accessed from websites listed or by contacting the corresponding author.

**Author contribution**

BH and JJ conceived the study and designed the *hybrid* assimilation methodology. JJ and AS performed the control and assimilation tests and carried out the data analysis. JJ prepared the manuscript with contributions from all BH and AS.

**Competing interests**

The authors declare that they have no conflict of interest.



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
