# Peer review of "How aerosol size matters in AOD assimilation and the optimization using Ångström exponent"

_Atmospheric Chemistry and Physics, 2022_

## Author Comment (AC1)

**Response to Referee #1**: We would like to thank the referee for the careful review and thoughtful suggestion, which helps us to improve the quality of the manuscript.

Our response follows (*the reviewer's comments are in italics and blue*)

*General Comments:*

*This manuscript explores the sensitivity of Ångström exponent to aerosol radius in a CTM, and uses the information together with satellite data to design a hybrid AOD assimilation. The topic is interesting to the aerosol and satellite committee. The paper is well written and the method is clearly descripted.*

*Major Comments:*

*My major concern is the authors attribute the discrepancy of Ångström exponent between satellite and model only to aerosol geometric mean radius. However, distribution of aerosol chemical compositions, aerosol mass density, refractive indices, relative humidity, hydroscopic growth, internal/external mixing in model assumptions all have uncertainties, some of them could be even more uncertain than aerosol size. Even we assume Ångström exponent is mostly sensitive to aerosol size distribution, the standard deviation of size distribution is also an important factor in additional to the radius. Are all these factors confirmed to have neglectable bias in the model, or their uncertainties hard to affect Ångström exponent? I would suggest add more analysis or at least discussion for this point.*

**Reply**: We agree with the referee that not only the aerosol size matters in the simulation of the aerosol optical property, but also the mentioned properties such as mixing ratio of aerosol species, refractive indices, and aerosol mass densities. Indeed, these all have an effect on the AOD/Ångström calculation. The Ångström values could there also be used to estimate other properties than aerosol sizes. For instance, Tsikerdekis et al. (2021) estimated the intensity and mass ratios of different aerosol species through simultaneously assimilating multiple optical observations, e.g., AOD and Ångström exponent.

A considerable amount of literatures on ground observations indicated there is remarkable spatial and temporal variation in the aerosol size distribution, with the geometric mean radius ranging from ten to several hundreds of nanometers (Costabile et al., 2009). It is therefore insufficient to describe the spatiotemporal variabilities using these fixed values as are used in practice. Otherwise, the model AOD and Ångström exponents are likely to be strongly biased and AOD assimilation result will be misled. Before designing hybrid assimilation (Ångström and AOD assimilation), we also made comparisons against other aerosol models (e.g., WRF-Chem and ECHAM-HAM) which indicated that our aerosol size assumptions are somewhat larger. As a first step to improve the AOD simulations by the model, the

mismatch between the simulated and observed Ångström exponent are attributed only to the error in the aerosol radii assumption, since these are uncertain for sure.

Concerning the standard deviation (stdv.) which is the other important parameter describe aerosol lognormal size distribution, the same choice was found in several other aerosol models. For instance, ECHAM-HAM (zhang et al., 2012) and EC-Earth3 (Noije et al., 2021) also use 1.59 and 2.0 for characterizing the stdv. of fine and coarse aerosol distribution, respectively. Similar choice (1.6 and 2.0) was also used in GEOS-Chem for describing their sulfate aerosols (Yu et al., 2009). Therefore, the stdv. of the size distribution is assumed to be more certain in this study.

We proposed the hybrid assimilation methodology that assimilates the Ångström (for estimating the most uncertain geometric mean radius) and AOD (for estimating aerosol concentration) sequentially. Validation against the independent measurements from the AERONET and ground $PM_{2.5}$ stations indicated that the estimated radii already brought the operator in model AOD simulation into better harmony with the assumption in satellite AOD retrieval algorithms.

Finally, our hybrid assimilation methodology is tested for the 4 days' case study in this study, and there is much space left for improvement in future. For instance, to optimizing the spatiotemporal variation and uncertainty in the standard deviation for aerosol size distribution, mass/number mixing ration of multiple aerosol species simultaneously, which might further bring the operator in model and satellite retrieval closer. The estimation of multiple uncertain parameters would benefit from extra observations, e.g., absorption aerosol optical depth.

To amplify that the aerosol size is only one of the uncertain parameters, extra lines have been added to page 15, line 16-29: "*A considerable amount of literatures on ground observations indicated that there is remarkable spatial and temporal variation in the aerosol size distribution, e.g., the geometric mean radius ranging from tens to hundreds of nanometers (Costabile et al., 2009). It is insufficient to describe these spatiotemporal varying characteristics using a fixed value as is used in practice. Using a fixed value, the model AOD and Ångström exponents are likely to be strongly biased as will be discussed in Section 4.2, and AOD assimilation result will be misled as is illustrated in Section 5.3. Meanwhile, comparisons against other aerosol models, e.g., WRF-Chem (Palacios-Peña et al., 2020) and ECHAM-HAM (Zhang et al., 2012a), indicated our aerosol size assumptions differ from them to some extent (mainly overestimated). Therefore, as a first step to improve the AOD simulations, we assign the mismatch between the simulated and observed Ångström exponents only to the errors in the assumption of aerosol radii in our Ångström assimilation.*

*The standard deviations (stdv.) are another key factor of the aerosol size distribution. The same choice as used in this study is present in several other aerosol models. For instance, ECHAM-HAM (Zhang et al., 2012a) and EC-Earth3 (van Noije et al., 2021) also use 1.59 and 2.0 for characterizing the stdv. of fine and coarse aerosol distribution, respectively. Similar choices (1.6 and 2.0) were also used in GEOS-Chem for describing sulfate aerosols (Yu and Luo, 2009). The stdv. of the size distribution is therefore assumed to be more certain in this study.*"

And in the ***Conclusions*** in page 26, line 33-35: "***Further optimization of this and other parameters, including spatiotemporal variability of the aerosol size distribution and mass ratio of different aerosol species, will be subject of future studies. A multiple parameter optimization would benefit from extra observations, e.g., absorption aerosol optical depth, in addition to the currently used Ångström and AOD. With such a multi-observation-assimilation it will be better possible to optimize aerosol mass concentrations including the different aerosol composition.***"

***Other comments:***

*Q1 The posterior radius could be evaluated with more measurements to make them more convincible. For instance, a brief comparison with size distribution from surface or aircraft observations in the literature.*

**Reply**: Thanks for the comments. The Ångström analysis in our hybrid assimilation aims to bring the aerosol radii in our LOTOS-EUROS closer to the assumption in the satellite retrieval algorithm which is not available. Aerosol size measurements could help to evaluate our Ångström analysis to some extent.

Details are added in page 21, line 4-11 "***Before validating our optimized LOTOS-EUROS AOD operator, these posterior aerosol radii obtained through Ångström analysis are first compared to some aerosol size measurements. As shown in Table 5 are several recent ground observations of aerosol lognormal size distribution over Europe. Compared to these observed radii, our prior assumption (0.35 μm) generally overestimated the fine aerosol size. The posterior geometric mean radii of the fine aerosols listed in Table 4 fall into the scope of most fine aerosol modes observed in Europe. However, as the documented aerosol size observations were collected in limited locations and different time period. They cannot fully represent the true aerosol radii or assumptions in the satellite retrieval algorithm in our simulation. Further evaluation will be carried out using the AERONET optical measurements and ground $PM_{2.5}$ observations collected synchronously.***"

**Table 5.** Summary of published observed lognormal size distribution parameters over Europe.

| Reference | observing location | time period | aerosol mode | geom. mean radius ($\mu$m) |
|---|---|---|---|---|
| Costabile et al. (2009) | 8 sites in/around Leipzig city | 2008-2009 | fine (accumulation) | 0.045-0.125 |
| Dall'Osto et al. (2019) | 3 sites in European high Arctic | 2013-2015 | fine (accumulation) | 0.04-0.05 |
| Wu and Boor (2021) | 314 sites in Europe | 1998-2017 | fine (accumulation) | 0.1-0.3 |
| Rose et al. (2021) | 39 global sites | summer in 2016/2017 | fine (mode 2) | 0.028-0.125 |
| Leinonen et al. (2022) | 21 sites in Europe and Arctic | 1996-2018 | fine (accumulation) | mostly from 0.075 to 0.1 |

*Q2 Any reason for the choices of wavelength pairs in Ångström exponent calculations (470-650nm for MODIS and 440-870nm for AERONET)? Would be the results different if choosing other wavelengths?*

**Reply**: At the two AERONET stations Cabauw and Leipzig used in this study, only Ångström$_{440\text{-}870}$ measurements are available as can be seen in the link *https://aeronet.gsfc.nasa.gov/cgibin/type_one_station_opera_v2_new*. Similarly, for the MODIS Deep Blue product over vegetated lands, only the Ångström$_{470\text{-}650}$ observations are available. To match these two observations, our LOTOS-EUROS simulated Ångström are interpolated at these two wavelength pairs 440-870 and 470-650 nm as well.

To make this clear, remarks are added in page 12, line 5-9 by saying "***AERONET AOD and Ångström observations are calculated at wavelengths (pairs) different from those used to interpolate the MODIS Deep Blue aerosol product; the details are presented in Table. 3. They are the only aerosol optical products released in the two AERONET stations (Cabauw and Leipzig) and MODIS Deep Blue product for vegetated lands. To accurately calculate AOD/Ångström difference between simulations and observations, our LOTOS-EUROS model simulated the AODs and Ångström at all these wavelengths (pairs).***"

*Q3 Section 4.2, paragraph 3. Modeled aerosol mass concentrations are underestimated. Depending on which chemical species are missing, this could largely affect Ångström exponent. Why do you think the bias of Ångström exponent is due to aerosol radius?*

**Reply**: See **Reply** to the ***Major comments.***

*Q4 Page 19, line 21. Please provide more evidence for assuming dust and sea salt are not important. What are the concentrations of dust and sea salt in your model simulation? Maybe this is also helpful to support your assumptions of their certain bins.*

**Reply**: Parameterized emissions for dust and sea salt aerosol were enabled in our model simulation. The former resulted in negligible dust levels over the 4 days' simulation. There were indeed some sea salt

aerosols, but most of them stayed over the ocean areas. These can be clearly seen in the following snapshots of the column concentration of total fine aerosol (PM$_{2.5}$), dust and sea salt. However, our assimilation was performed only in the MODIS Deep Blue observational space, which only provides measurements over land areas. Therefore, both sea salt and dust aerosols have limited effects on our assimilation.

On the other hand, our prior simulation as seen in Fig. 2d already underestimated the Ångström exponents compared to observations shown in Fig. 2b. The mixture of coarse sea salt and dust aerosols would lead to a further low Ångström exponent simulation. This gives us confidence to assume that dust and sea salt aerosols have no effect on our assimilation.

To make this clearer, remarks is now added in page 20, line 7-11: "***During the test period, our LOTOS-EUROS simulated negligible dust levels. There were indeed some sea salt aerosols, but most of them stay over the ocean areas. This can be clearly seen in the snapshots of the column concentration of fine aerosol, dust and sea salt in Fig. S6. However, our AOD assimilation was performed only in the MODIS Deep Blue observational space as has been illustrated in Section 5.1.1, which only provides measurements over land areas. Therefore, both sea salt and dust aerosols have limited effects on our assimilation, they are assumed certain and not estimated in this study.***"

[Figure]

*Figure S6. Snapshots of the LOTOS-EUROS simulated column concentration of the fine aerosol (a), dust (b) and sea salt (c) at 2012 July 24, 10:00 to 11:00.*

*Q6 Fig 7-9: maybe narrow the range of colorbars to make the distribution clearer.*

**Reply**: We have revised the Fig.2 (in page 11), Fig. 7 (in page 22), Fig. 8 (in page 23) and Fig. 9 (in page 24) with smaller AOD/PM$_{2.5}$ range as well as higher figure resolutions as can be seen in the revised manuscript. This resulted in clearer distributions.

**New Reference**

1. Wu, T. and Boor, B. E.: Urban aerosol size distributions: a global perspective, Atmospheric Chemistry and Physics, 21, 8883–8914, https://doi.org/10.5194/acp-21-8883-2021, https://acp.copernicus.org/articles/21/8883/2021/, 2021.

2. Dall'Osto, M., Beddows, D. C. S., Tunved, P., Harrison, R. M., Lupi, A., Vitale, V., Becagli, S., Traversi, R., Park, K.-T., Yoon, Y. J., Massling, A., Skov, H., Lange, R., Strom, J., and Krejci, R.: Simultaneous measurements of aerosol size distributions at three sites in the European high Arctic, Atmospheric Chemistry and Physics, 19, 7377–7395, https://doi.org/10.5194/acp-19-7377-2019, https://acp.copernicus.org/articles/19/7377/2019/, 2019.

3. Palacios-Peña, L., Fast, J. D., Pravia-Sarabia, E., and Jiménez-Guerrero, P.: Sensitivity of aerosol optical properties to the aerosol size distribution over central Europe and the Mediterranean Basin using the WRF-Chem v.3.9.1.1 coupled model, Geoscientific Model Development,13, 5897–5915, https://doi.org/10.5194/gmd-13-5897-2020, https://gmd.copernicus.org/articles/13/5897/2020/, 2020.

4. Costabile, F., Birmili, W., Klose, S., Tuch, T., Wehner, B., Wiedensohler, A., Franck, U., König, K., and Sonntag, A.: Spatio-temporal variability and principal components of the particle number size distribution in an urban atmosphere, Atmospheric Chemistry and Physics, 9, 3163–3195, https://doi.org/10.5194/acp-9-3163-2009, https://acp.copernicus.org/articles/9/3163/2009/, 2009.

5. Leinonen, V., Kokkola, H., Yli-Juuti, T., Mielonen, T., Kühn, T., Nieminen, T., Heikkinen, S., Miinalainen, T., Bergman, T., Carslaw, K., Decesari, S., Fiebig, M., Hussein, T., Kivekäs, N., Krejci, R., Kulmala, M., Leskinen, A., Massling, A., Mihalopoulos, N., Mulcahy, J. P., Noe, S. M., van Noije, T., O'Connor, F. M., O'Dowd, C., Olivie, D., Pernov, J. B., Petäjä, T., Seland, Ø., Schulz, M., Scott, C. E., Skov, H., Swietlicki, E., Tuch, T., Wiedensohler, A., Virtanen, A., and Mikkonen, S.: Comparison of particle number size distribution trends in ground measurements and climate models, Atmospheric Chemistry and Physics, 22, 12 873–12 905, https://doi.org/10.5194/acp-22-12873-2022, https://acp.copernicus.org/articles/22/12873/2022/, 2022

6. Rose, C., Collaud Coen, M., Andrews, E.: Seasonality of the particle number concentration and size distribution: a global analysis retrieved from the network of Global Atmosphere Watch (GAW) near-surface observatories, Atmospheric Chemistry and Physics, 21, 17 185–17 223, https://doi.org/10.5194/acp-21-17185-2021, https://acp.copernicus.org/articles/21/17185/2021/, 2021.

7. Zhang, K., O'Donnell, D., Kazil, J., Stier, P., Kinne, S., Lohmann, U., Ferrachat, S., Croft, B., Quaas, J., Wan, H., Rast, S., and Feichter, J.: The global aerosol-climate model ECHAM-HAM, version 2: sensitivity to improvements in process representations, Atmos. Chem. Phys., 12, 8911–8949, https://doi.org/10.5194/acp-12-8911-2012, 2012.

8. Yu, F. and Luo, G.: Simulation of particle size distribution with a global aerosol model: contribution of nucleation to aerosol and CCN number concentrations, Atmospheric Chemistry and Physics, 9, 7691–7710, https://doi.org/10.5194/acp-9-7691-2009, https://acp.copernicus.org/articles/9/7691/2009/, 2009.

9. Tsikerdekis, A., Schutgens, N. A. J., and Hasekamp, O. P.: Assimilating aerosol optical properties related to size and absorption from POLDER/PARASOL with an ensemble data assimilation system, Atmos. Chem. Phys., 21, 2637–2674, https://doi.org/10.5194/acp-21-2637-2021, 2021.

10. van Noije, T., Bergman, T., Le Sager, P., O'Donnell, D., Makkonen, R., Gonçalves-Ageitos, M., Döscher, R., Fladrich, U., von Hardenberg, J., Keskinen, J.-P., Korhonen, H., Laakso, A., Myriokefalitakis, S., Ollinaho, P., Pérez García-Pando, C., Reerink, T., Schrödner, R., Wyser, K., and Yang, S.: EC-Earth3-AerChem: a global climate model with interactive aerosols and atmospheric chemistry participating in CMIP6 , Geosci. Model Dev., 14, 5637–5668, https://doi.org/10.5194/gmd-14-5637-2021, 2021.

---

## Author Comment (AC2)

**Response to Referee #2**: We would like to thank the referee for the careful review and insightful suggestion throughout the manuscript, which helps us to improve the quality of the manuscript.

Our response follows *(the reviewer's comments are in italics and blue)*

***General Comments:***

*This study attempts to investigate the effects of sequential assimilation of satellite-based aerosol size information (i.e., À ngstrò m exponents) and aerosol optical depths (AOD) on the analyse of the aerosol concentrations. The assimilation experiments are conducted over the European region with the MODIS Deep Blue products. The results demonstrate that the assimilation of the MODIS observed aerosol size information could improve the surface fine particles analyses by correcting the model assumed aerosol geometric radius and subsequent the AOD observation operator. The paper is generally well written and scientific sound.*

Main comments:

*Q1. It looks the simulated À ngstrò m exponents without any data assimilation are too low (below zero). The authours also claim that there are no dust events during the studied period, so does this mean that the default parameters of the aerosol radii for the fine aerosol particles such as the sulfate or carbonaces aerosol are too large in the LOTOS-EUROS model. If this is ture, why the model uses those values.*

**Reply**: The description of aerosol radii in the model is indeed very important. A considerable amount of literatures on ground observations indicated that there is remarkable spatial and temporal variation in the aerosol size distribution, with the geometric mean radius ranging from ten to several hundreds of nanometers (Costabile et al., 2009). It is difficult to describe the spatiotemporal varying features using fixed values like most models do in practice. In addition, also the Ångström exponent measurements are uncertain, and might change with the assumption of aerosol size distribution in different satellite retrieval product. Our hybrid assimilation is therefore designed to solve this issue. Different from a standard AOD assimilation that directly assimilates AOD observations and ignores the potential mismatch of the particle radius distribution, the hybrid approach first estimates suitable aerosol size parameters by assimilating Ångström exponent observations, before performing the AOD assimilation.

To describe this better in the manuscript, remarks are now added in page 15, line 16-20: "***A considerable amount of literatures on ground observations indicated that there is remarkable spatial and temporal variation in the aerosol size distribution, e.g., the geometric mean radius ranging from tens to***

*hundreds of nanometers (Costabile et al., 2009). It is insufficient to describe these spatiotemporal varying characteristics using a fixed value as is used in practice. Using a fixed value, the model AOD and Ångström exponents are likely to be strongly biased as will be discussed in Section 4.2, and AOD assimilation result will be misled as is illustrated in Section 5.3.*"

*2. The results in Figure 2 demonstrate that there are some too low AÌ ngstroÌ m exponents. Probably, the quality of the satellite retrievals of the AÌ ngstroÌ m exponent over such region is not good. How does this unusual observation affect your assimilation result?*

**Reply**: There are indeed some inconsistent MODIS Ångström measurements assimilated as shown in green colored box in Fig. 2b. However, the aerosol radii are assumed to be spatially and temporally constant during the whole assimilation window. The radii would be nudged to fit the dominant MODIS Ångström exponents while less influenced by these few inconsistent data as present. Of course, data quality control should be introduced if the spatial variability of aerosol radii is explored in future work.

To make this clear, remarks are added in page 19, line 27-30: "*The aerosol radii are assumed to be spatially and temporally constant during the short period used for the experiments. Spatially varying radii would of course allow the assimilated Ångström exponent to better fit the MODIS Ångström exponent. However, the locally inconsistent MODIS Ångström observations found during comparison with AERONET observations in Section 3.2 would introduce strong local mis-adjustments in case a (large) spatial degree of freedom is allowed. Data quality control for excluding these polluted data is required. Introducing spatial variations also requires information on spatial correlations and would increase computational costs; hence, this aspect has not been explored in this study.*"

*3. As the assumption of diagonal matrix of the model background covariance B for AOD, do you mean only the aerosol mass concentrations over the model grid with MODIS observation could be optimized? How about the model grids without any available observations to be assimilated? Does this induce some unreason aerosol distributions?*

**Reply**: Thanks for the in-depth comment. Yes, we only performed the *AOD analysis* over the pix where MODIS AOD is available.

In data assimilation, all states ($\in R^n$) could be optimized by the observations correlated. The spatial correlation (anisotropic) matrix is described using a background covariance matrix $\mathbf{B}$ ($\in R^{n*n}$) in

variational method, or using an ensemble approximated **B** ($\in$ R$^{n*n}$, but much lower rank) in EnKF. In practice, how observations would help improve the state estimation not only depends on the observations, but also depends on the spatial correlation in **B**. For this study, we only aimed to explore how AOD measurements would help/harm the state estimation. To prevent the influence from spatial correlation, we carried out the assimilation in the subspace where MODIS measurements are available and assumed the **B** is independent.

To make this clear, remarks are added in page 19, line 8-12: "***In the background covariance $B_\tau$, the main diagonal defines the assumed variance of the model AOD, while the off-line elements represent the correlations between two AOD values in different grid cells. In this study the focus is on using the available AOD observations to obtain insight in the validity of the assumptions on the aerosol size distribution. Correlations between grid cells that would also influence the assimilation in practice are therefore simply ignored, and all optimizations are done per grid cell.***"

*4. P19 Line 29, How to obtain the optimal aerosol radius using a 4DvEnvar? Please clarify it more detail?*

**Reply**: Details of the 4DEnVar are now added in the ***Supplementary material***.

Remarks are added in page 20, line 18-19: "***The aerosol radius ra that minimizes the cost function Eq. (12) is obtained using a 4DvEnvar method (Liu et al., 2008) and the detailed procedures can be found in the Ångström analysis cost function minimization.***"

**2  Ångström analysis cost function minimization**

The minimization of the cost function follows the 4DEnVar processes. An ensemble of aerosol radius vector are generated randomly using the prior $\boldsymbol{r}_b$ and the assumed error covariance $\mathbf{B}_r$:

$$[\, \boldsymbol{r}_1, ..., \boldsymbol{r}_N \,] \tag{1}$$

An ensemble of Ångström model simulations then forward with the ensemble aerosol radius vectors in parallel:

$$[\, \mathcal{M}(\boldsymbol{r}_1), ..., \mathcal{M}(\boldsymbol{r}_N) \,] \tag{2}$$

Denote the ensemble perturbation matrix by:

$$\mathbf{L}' = \frac{1}{\sqrt{N-1}} [\, \boldsymbol{r}_1 - \bar{\boldsymbol{r}}, \,...,\, \boldsymbol{r}_N - \bar{\boldsymbol{r}} \,] \tag{3}$$

and mean of ensemble simulation by:

$$\overline{\mathcal{M}(\boldsymbol{r})} = \frac{1}{N} \sum_{i=1}^{N} \mathcal{M}(\boldsymbol{r}_i) \tag{4}$$

where $\bar{\boldsymbol{r}}$ is the mean of the ensemble aerosol radii. In the 4DEnVar assimilation algorithm, the optimal radii $\boldsymbol{r}_a$ is defined as weighted sum of the columns of the perturbation matrix $\mathbf{L}'$ using weights from a control variable vector $\boldsymbol{w}$:

$$\boldsymbol{r}_a = \bar{\boldsymbol{r}} + \mathbf{L}' \boldsymbol{w} \tag{5}$$

The cost function could then be reformulated as:

$$\mathcal{J}(\boldsymbol{w}) = \frac{1}{2} \boldsymbol{w}^{\mathsf{T}} \boldsymbol{w} \;+\; \frac{1}{2} \{\, \mathbf{M}\mathbf{L}'\boldsymbol{w} + \overline{\mathcal{M}(\boldsymbol{r})} - \boldsymbol{\mathcal{A}} \,\}^{\mathsf{T}} \mathbf{R}_{\mathcal{A}}^{-1} \{\, \mathbf{M}\mathbf{L}'\boldsymbol{w} + \overline{\mathcal{M}(\boldsymbol{r})} - \boldsymbol{\mathcal{A}} \,\} \tag{6}$$

here $\mathbf{M}$ is the linearization of the LOTOS-EUROS Ångström simulation model required for cost function minimization, and is approximated by:

$$\mathbf{M}\mathbf{L}' \approx \frac{1}{\sqrt{N-1}} [\, \mathcal{M}(\boldsymbol{r}_1) - \overline{\mathcal{M}(\boldsymbol{r})}, \,...,\, \mathcal{M}(\boldsymbol{r}_N) - \overline{\mathcal{M}(\boldsymbol{r})} \,] \tag{7}$$

with the uncertainty in radii transferred into the observations space, the minimum of the cost function in Eq. 6 could then be directly calculated, and the posterior emission $\boldsymbol{r}_a$ subsequently be updated.